# OpenCap: Human movement dynamics from smartphone videos

**Scott D. Uhlrich**[1,☯]*, **Antoine Falisse**[1,☯]*, **Łukasz Kidziński**[1,☯], **Julie Muccini**[2], **Michael Ko**[2], **Akshay S. Chaudhari**[2,3], **Jennifer L. Hicks**[1], **Scott L. Delp**[1,4,5]

**1** Departments of Bioengineering, Stanford University, Stanford, California, United States of America, **2** Radiology, Stanford University, Stanford, California, United States of America, **3** Biomedical Data Science, Stanford University, Stanford, California, United States of America, **4** Mechanical Engineering, Stanford University, Stanford, California, United States of America, **5** Orthopaedic Surgery, Stanford University, Stanford, California, United States of America

☯ These authors contributed equally to this work.
* suhlrich@stanford.edu (SDU); afalisse@stanford.edu (AF)

## Abstract

Measures of human movement dynamics can predict outcomes like injury risk or musculo-skeletal disease progression. However, these measures are rarely quantified in large-scale research studies or clinical practice due to the prohibitive cost, time, and expertise required. Here we present and validate OpenCap, an open-source platform for computing both the kinematics (i.e., motion) and dynamics (i.e., forces) of human movement using videos captured from two or more smartphones. OpenCap leverages pose estimation algorithms to identify body landmarks from videos; deep learning and biomechanical models to estimate three-dimensional kinematics; and physics-based simulations to estimate muscle activations and musculoskeletal dynamics. OpenCap's web application enables users to collect synchronous videos and visualize movement data that is automatically processed in the cloud, thereby eliminating the need for specialized hardware, software, and expertise. We show that OpenCap accurately predicts dynamic measures, like muscle activations, joint loads, and joint moments, which can be used to screen for disease risk, evaluate intervention efficacy, assess between-group movement differences, and inform rehabilitation decisions. Additionally, we demonstrate OpenCap's practical utility through a 100-subject field study, where a clinician using OpenCap estimated musculoskeletal dynamics 25 times faster than a laboratory-based approach at less than 1% of the cost. By democratizing access to human movement analysis, OpenCap can accelerate the incorporation of bio-mechanical metrics into large-scale research studies, clinical trials, and clinical practice.

## Author summary

Analyzing how humans move, how we coordinate our muscles, and what forces act on the musculoskeletal system is important for studying neuro-musculoskeletal conditions. Traditionally, measuring these quantities requires expensive laboratory equipment, a trained expert, and hours of analysis. Thus, high-quality measures of human movement are rarely

**Data Availability Statement:** The data from the laboratory validation and field studies are available at https://simtk.org/projects/opencap. Identifiable video data is shared for the laboratory validation study. Participants whose identifiable videos are

shared provided written consent to the release of these videos. The source code is available at https://github.com/stanfordnmbl/opencap-core (archived in Zenodo: https://doi.org/10.5281/zenodo.7419967) and https://github.com/stanfordnmbl/opencap-processing (archived in Zenodo: https://doi.org/10.5281/zenodo.7419973) under a permissive open-source license (Apache 2.0), and the software can be used through our web and mobile applications by visiting https://opencap.ai. These applications and the cloud computing that enables them will be freely available to the research community for the foreseeable future.

**Funding:** SDU, AF, LK, JM, ASC, JLH, and SLD were supported by the National Institutes of Health (https://www.nih.gov; grant 1P41EB027060-01A1) and the Wu Tsai Human Performance Alliance (https://humanperformancealliance.org). ASC and MK were supported by Philips Healthcare (https://www.usa.philips.com/healthcare) and the National Institutes of Health (https://www.nih.gov; grant 1R01AR077604-01). The funders had no role in study design, data collection and analysis, decision to publish, or preparation of the manuscript.

incorporated into clinical practice and large-scale research studies. The advent of computer vision methods for locating human joints from standard videos offers a promising alternative to laboratory-based movement analysis. However, it is unclear whether these methods provide sufficient information for informing biomedical research and clinical practice. Here, we introduce OpenCap, an open-source, web-based software tool for computing the motion (e.g., joint angles) and the musculoskeletal forces underlying human movement (e.g., joint forces) from smartphone videos. OpenCap combines advances in computer vision, machine learning, and musculoskeletal simulation to make movement analysis widely available without specialized hardware, software, or expertise. We validate OpenCap against laboratory-based measurements and show its usefulness for applications including screening for disease risk, evaluating intervention efficacy, and informing rehabilitation decisions. Finally, we highlight how OpenCap enables large-scale human studies of human movement in real-world settings.

## Introduction

Evaluating the dynamics (i.e., musculoskeletal forces) and control of human movement is important for understanding and managing musculoskeletal and neuromuscular diseases. For example, the loading in osteoarthritic joints predicts osteoarthritis progression [1], the distribution of moments generated by muscles about lower-extremity joints when rising from a chair relates to falling in older adults [2–4], and the between-limb asymmetry of muscle and ground reaction forces while performing demanding tasks relates to functional outcomes after joint surgery [5–7]. Despite their utility, metrics of movement dynamics are rarely measured in clinical practice. Instead, visual movement evaluations or general functional tests that require basic instruments, like a stopwatch or goniometer, are used to inform clinical decisions and as outcomes for clinical trials.

The quantitative analysis of movement dynamics can provide deeper and more reproducible insights than visual evaluations and simple functional tests; however, this analysis is resource intensive, which has impeded its use in large-scale studies and clinical practice. Traditionally, motion analysis requires a fixed lab space with more than $150,000 of equipment (Fig 1, top row). Kinematics (e.g., joint angles) are measured with a marker-based motion capture system that uses eight or more specialized cameras to capture the three-dimensional (3D) trajectories of markers placed on a subject. Dynamic measures (e.g., joint moments and powers), also referred to as kinetics, can be estimated with the additional measurement of ground reaction forces from force plates mounted beneath the floor. Musculoskeletal modeling and simulation tools [8–10] combine measures of kinematics, kinetics, and muscle activation from electromyography to enable deeper investigations of motor control and musculoskeletal loading (e.g., muscle coordination and joint forces). This comprehensive analysis of movement is infrequently used outside of small-scale research studies because collecting data on a single participant, processing it, and, optionally, generating dynamic musculoskeletal simulations typically takes several days for a trained expert.

Studies of movement dynamics with hundreds of participants have elucidated biomechanical markers that predict injury risk or surgical outcomes [11–13]. However, studies of this scale are expensive and rare—the median number of subjects included in biomechanics studies is between 12 and 21 [14,15]. There is a need for inexpensive, scalable, and accurate tools for estimating movement dynamics on orders of magnitude more individuals in their natural environments. Modern data science techniques could then leverage these large datasets to

## Mocap: marker-based motion analysis

## OpenCap: video-based motion analysis

**Fig 1. Marker-based motion capture (Mocap) versus video-based (OpenCap) analysis of human movement dynamics. (Top row)** Marker-based movement analysis usually occurs in a motion capture laboratory, and a comprehensive study of musculoskeletal dynamics typically requires more than two days of an expert's time and equipment worth more than $150,000. **(Bottom row)** OpenCap enables the study of musculoskeletal dynamics in less than 10 minutes of hands-on time and with equipment worth less than $700 (assuming users need to purchase new mobile devices). OpenCap can be used anywhere with internet access and requires a minimum of two iOS devices (e.g., iPhones or iPads). **(Right panel)** OpenCap enables the estimation of kinematic, dynamic, and musculotendon parameters, many of which were previously only accessible using marker-based motion capture and force plate analysis.

explore the role of movement in health and disease, facilitating the identification and clinical translation of quantitative movement biomarkers.

Mobile tools for estimating kinematics have been developed, but most are still too expensive and time consuming for large-scale research studies and clinical translation, and none enable full-body analysis of movement dynamics. Inertial measurement units, the most widely used of these tools, can accurately estimate kinematics [16], but commercially available sensors remain expensive, time-consuming to don and doff, and utilize proprietary algorithms. Recent advances in physics-based simulation of the musculoskeletal system have estimated dynamics from inertial measurement unit–based motion capture [17,18], but these algorithms are not publicly available and have not been translated beyond small-scale feasibility studies.

Measuring kinematics with video cameras is another promising approach made possible by recent advancements in human pose estimation algorithms [19,20]. Open-source, two-dimensional (2D) pose estimation algorithms (e.g., OpenPose [21]) have enabled 2D kinematic analyses [22] and can generate inputs for machine learning models that predict kinematic and dynamic measures [23,24]. While these machine learning models are useful for specific applications, they may not generalize to other measures, tasks, and populations not represented in their training data. Another potentially more generalizable approach is to triangulate the body keypoints (e.g., joint centers) identified by pose estimation algorithms on multiple videos [25–30] and track these 3D positions with a musculoskeletal model and physics-based simulation. However, the sparse set of 3D keypoints identified by these algorithms does not fully characterize the translations and rotations of all body segments; thus, it is unclear whether these keypoints are expressive and accurate enough to inform movement research. Commercial markerless motion capture systems accurately estimate kinematics [31], but they typically

require many wired cameras, proprietary software, and specialized computing resources. The ubiquity of smartphone cameras could enable video-based motion capture without the need to purchase specialized equipment, but it is unclear whether kinematics can be accurately estimated from a small number of devices that lack hardware synchronization. If the challenges of computing accurate kinematics and dynamics from smartphone video can be addressed, smartphone-based analysis of musculoskeletal dynamics has the potential to overcome the translational barriers faced by current movement analysis technologies.

Here we introduce OpenCap, open-source, web-based software that is freely available to the research community for estimating the 3D kinematics and dynamics of human movement from videos captured with two or more smartphones (Fig 1, bottom row). OpenCap brings together decades of advances in computer vision and musculoskeletal simulation to make the analysis of movement dynamics available without specialized hardware, software, or expertise. We first validate kinematic and dynamic measures estimated with OpenCap against gold standard measures computed with marker-based motion capture and force plates. Next, we explore whether OpenCap estimates dynamic measures with sufficient accuracy to be used for disease risk screening, evaluating intervention efficacy, studying between-group movement differences, and tracking rehabilitation progress. After validating these measures in the laboratory, we highlight how OpenCap enables clinicians to measure movement dynamics in large cohorts in real-world settings.

## Results

### Data collection with OpenCap

Setting up a data collection with OpenCap takes under five minutes and requires two iOS devices (iPhone, iPad, or iPod), two tripods, a calibration checkerboard (printed with a standard printer), and another device to run OpenCap's web application (e.g., a laptop). After pairing the iOS devices to the web application, users are guided through camera calibration, data collection, and visualization of 3D kinematics. Kinematics are estimated from video using deep learning models and inverse kinematics in OpenSim [8,10], and dynamics are estimated using a physics-based musculoskeletal simulation approach (Fig 2, S1 Movie). OpenCap leverages cloud computing for data processing using a scalable server architecture.

### Validation against the lab-based gold standard

We validated OpenCap using two iPhones against marker-based motion capture and force plate analysis in a cohort of ten healthy individuals for several activities (walking, squatting, rising from a chair, and drop jumps). OpenCap estimated joint angles with a mean absolute error (MAE) of 4.5˚, ground reaction forces with an MAE of 6.2% bodyweight, and joint moments with an MAE of 1.2% bodyweight*height (Table 1; additional validation in Methods: Validation; S1–S4 Tables and Figs A-L in S1 Appendix).

### Disease risk screening: Knee loading during walking

We then explored whether OpenCap is sufficiently accurate to estimate measures of joint loading that could be used to screen for individuals at risk of rapid progression of medial knee osteoarthritis and to evaluate the efficacy of a non-surgical intervention. We first evaluated how accurately OpenCap estimates the early-stance peak knee adduction moment, which predicts rapid progression of medial knee osteoarthritis [1]. The ten healthy individuals walked naturally (i.e., with a self-selected strategy) and with a trunk sway gait modification that typically reduces the knee adduction moment [32]. OpenCap predicted the early-stance peak knee

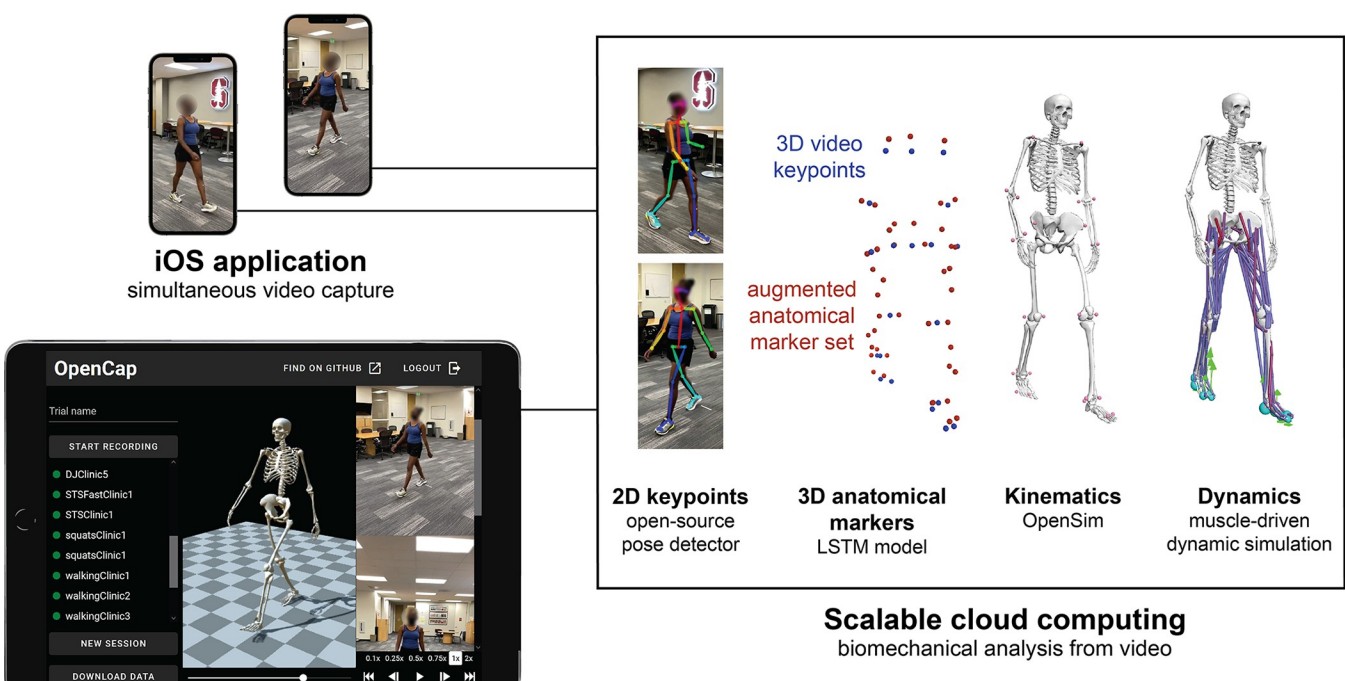

**Fig 2. OpenCap comprises a smartphone application, a web application, and cloud computing.** To collect data, users open an application on two or more iOS devices and pair them with the OpenCap web application. The web application enables users to record videos simultaneously on the iOS devices and to visualize the resulting 3-dimensional (3D) kinematics. In the cloud, 2D keypoints are extracted from multi-view videos using open-source pose estimation algorithms. The videos are time synchronized using cross-correlations of keypoint velocities, and 3D keypoints are computed by triangulating these synchronized 2D keypoints. These 3D keypoints are converted into a more comprehensive 3D anatomical marker set using a recurrent neural network (LSTM) trained on a large motion capture dataset. 3D kinematics are then computed from marker trajectories using inverse kinematics and a musculoskeletal model with biomechanical constraints. Finally, dynamic measures are estimated using muscle-driven dynamic simulations that track 3D kinematics.

**Table 1. Mean absolute error (MAE) in kinematics and kinetics from OpenCap compared to laboratory-based motion capture and force plates.**

| Kinematics (MAE) | Walking | Squat | Sit-to-stand | Drop jump | Mean |
|---|---|---|---|---|---|
| Rotations (n = 18) [°] | 4.1 (2.3–6.6) | 4.1 (1.8–7.2) | 4.7 (1.7–10.3) | 5.1 (2.3–8.6) | 4.5 |
| Translations (n = 3) [mm] | 12.3 (6.8–19.6) | 12.3 (5.8–18.4) | 13.2 (5–20.3) | 11.5 (6.3–16.5) | 12.3 |
| **Ground reaction forces (MAE)** | | | | | |
| Vertical [%BW] | 8.2 (7.5%) | 6.4 (20.0%) | 5.7 (13.4%) | 25.2 (13.8%) | 11.4 (13.7%) |
| Anterior-posterior [%BW] | 2.1 (6.7%) | 1.3 (37.5%) | 1.9 (31.0%) | 8.9 (17.3%) | 3.5 (23.1%) |
| Medio-lateral [%BW] | 1.1 (17.1%) | 5.7 (85.4%) | 3.2 (110.5%) | 5.3 (29%) | 3.8 (60.5%) |
| **Joint moments (MAE)** | | | | | |
| All degrees of freedom (n = 15) [%BW*ht] | 0.75 (0.20–1.32, 19%) | 0.97 (0.11–1.93, 45%) | 0.68 (0.13–1.09, 60%) | 2.50 (1.15–5.90, 25%) | 1.22 (37%) |

Errors for each activity were averaged over trials and participants (n = 10), and the reported mean is an average over activities and degrees of freedom (six for pelvis position and orientation [kinematics only], three for the lumbar, three per hip, one per knee, and two per ankle). Forces are expressed in percent bodyweight (BW) and moments in percent BW times height (ht). Kinematic and joint moment errors are presented as the mean and range over the degrees of freedom, and kinetic errors are additionally presented as the MAE as a percentage of the range. Root mean squared error in kinematics and kinetics are available in S2–S4 Tables. Average kinematic, ground reaction force, and joint moment waveforms estimated using OpenCap and Mocap are presented in Figs A–L in S1 Appendix.

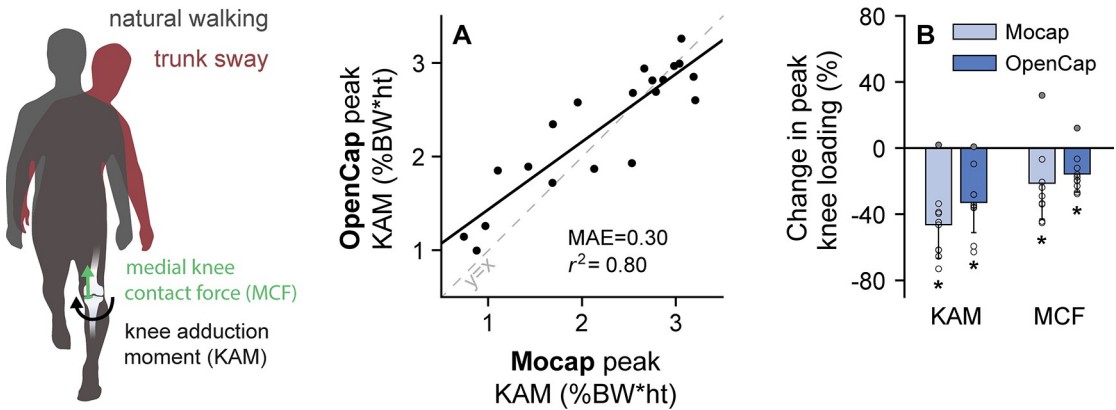

**Fig 3. Medial knee loading during walking.** We evaluated how accurately OpenCap estimates the knee adduction moment (KAM), a measure of medial knee loading that predicts knee osteoarthritis progression, and how knee loading changes with a modified walking pattern. Participants (n = 10) walked naturally and with a trunk sway gait modification. **(A)** OpenCap estimated the early-stance peak KAM with $r^2 = 0.80$, compared to an analysis using marker-based motion capture and force plates (Mocap). The KAM is normalized by bodyweight (BW) and height (ht). **(B)** The mean (bar) and standard deviation (error bar) across participants (circles) are shown for the changes in the peak KAM and peak medial contact force (MCF), which is a more comprehensive measure of medial knee loading, from natural to trunk sway walking (*$P < .05$). OpenCap detected the reductions in peak KAM and MCF ($P < .006$, t test and Wilcoxon signed rank test) that were measured with Mocap ($P < .016$, t tests). Furthermore, OpenCap correctly identified the one individual who did not reduce KAM or MCF as estimated by Mocap (filled circles).

adduction moment with $r^2 = 0.80$ ($r$: Pearson correlation coefficient) and an MAE of 0.30% bodyweight*height compared to marker-based motion capture and force plates (Fig 3). This error is smaller than a range of thresholds for detecting knee osteoarthritis symptoms and progression (0.5–2.2% bodyweight*height [1,33–35]). We then evaluated whether OpenCap could estimate changes induced by the trunk sway modification in the peak knee adduction moment as well as the peak medial contact force, which is a more comprehensive loading metric that is often targeted by knee osteoarthritis interventions [36,37]. At the group level, OpenCap captured expected reductions in the early-stance peak knee adduction moment and peak medial contact force from the trunk sway gait modification (16–33% reductions, $P < .006$; t test and Wilcoxon signed rank test, n = 10, Fig 3B). Significant changes in the same direction were also detected with motion capture and force plates (21–46% reductions, $P < .016$; t tests, n = 10); further details about these statistical tests can be found in Table A in S1 Appendix. For this sample size, OpenCap had a 92% chance (post-hoc power averaged across tests) of detecting these expected group differences at the significance level alpha = .05, compared to the 77% chance from motion capture and force plates. At the individual level, OpenCap correctly predicted the directional change in both peak loading measures (decrease for nine individuals and increase for one individual) induced by trunk sway. OpenCap's ability to accurately estimate knee loading and changes in loading during walking suggests that it could be used to identify individuals with medial knee osteoarthritis who may be at risk of rapid disease progression and to evaluate the effect of a gait modification on individual and group levels [38,39].

## Detecting between-group differences: Kinetic differences during sit-to-stand

We then explored whether OpenCap is useful for studying differences in movement dynamics that commonly exist between young and older adults. Strategies for rising from a chair vary with age and are associated with different muscle force requirements [2]. Older adults often use a rising strategy with increased trunk flexion, which shifts the muscular demand from the

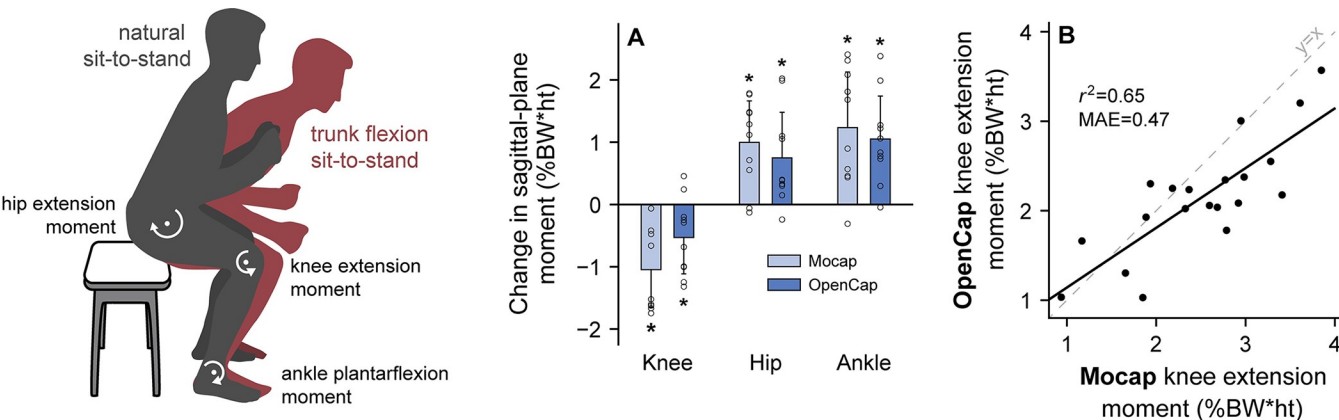

**Fig 4. Distribution of lower-extremity joint moments when rising from a chair.** To evaluate OpenCap's ability to detect between-group differences in dynamics, we computed differences in lower-extremity joint moments while rising from a chair that commonly exist between young and older adults. Individuals (n = 10) stood naturally and with increased trunk flexion, a strategy used by individuals with knee extensor weakness that shifts muscle demand to the hip extensors and ankle plantarflexors. **(A)** The mean (bar) and standard deviation (error bar) across participants (open circles) are shown for the changes in knee extension, hip extension, and ankle plantarflexion moments, averaged over the rising phase, from the natural to trunk flexion condition (*$P < .05$). Moments are normalized to percent bodyweight (BW) times height (ht). OpenCap identified the changes in joint moments ($P = .004–.024$, t tests) that were identified with motion capture and force plates (Mocap, $P = .001–.002$, t tests). **(B)** The rising-phase-averaged knee extension moment values for each participant and condition are shown. OpenCap estimated the knee extension moment with $r^2 = 0.65$ compared to simulations that used motion capture and force plate data as input (Mocap).

knee extensors to the hip extensors and ankle plantarflexors [40]; this strategy is associated with low functional muscle strength [4], which relates to fall risk [3]. We simulated differences in rising strategies between age groups by instructing ten healthy individuals to rise from a chair five times naturally, then five times with increased trunk flexion (Fig 4). At the group level, OpenCap estimated the expected reduction in the knee extension moment ($P = .024$, t test, n = 10) and increase in the hip extension ($P = .020$, t test, n = 10) and ankle plantarflexion moments ($P = .004$, t test, n = 10), averaged over the rising phase, from the natural to the increased trunk flexion condition. The direction of these changes matched what was measured with motion capture and force plates ($P = .002–.003$, t tests, n = 10); further details about these statistical tests can be found in Table B in S1 Appendix. For this sample size, OpenCap had a 65% chance (post-hoc power averaged across tests) of detecting these expected between-condition differences at the significance level alpha = .05, compared to the 89% chance from motion capture and force plates. OpenCap also predicted the peak knee extension moment with $r^2 = 0.65$ and MAE = 5.5Nm (0.47%bodyweight*height) compared to marker-based motion capture and force plates. This error is similar to the average loss in strength that occurs over six years in middle-aged adults (0.93Nm/year [41], see S1 Appendix for further discussion). Together, these findings suggest that OpenCap can be used to study differences in movement dynamics between young and older adults and can identify individuals with low knee extensor strength who may benefit from muscle strengthening interventions [2].

## Informing rehabilitation: Muscle activation asymmetry during squatting

Finally, we explored whether OpenCap can accurately estimate measures of muscle force associated with rehabilitation progress. Restoring between-limb symmetry in knee extensor muscle force generation is often a goal of rehabilitation following knee surgeries, and identifying persistent asymmetry prior to rehabilitation discharge can prevent poor functional outcomes [5,6,42]. To simulate post-surgical asymmetries, we instructed the ten healthy individuals to perform five squats naturally, then asymmetrically by reducing the force under their left foot

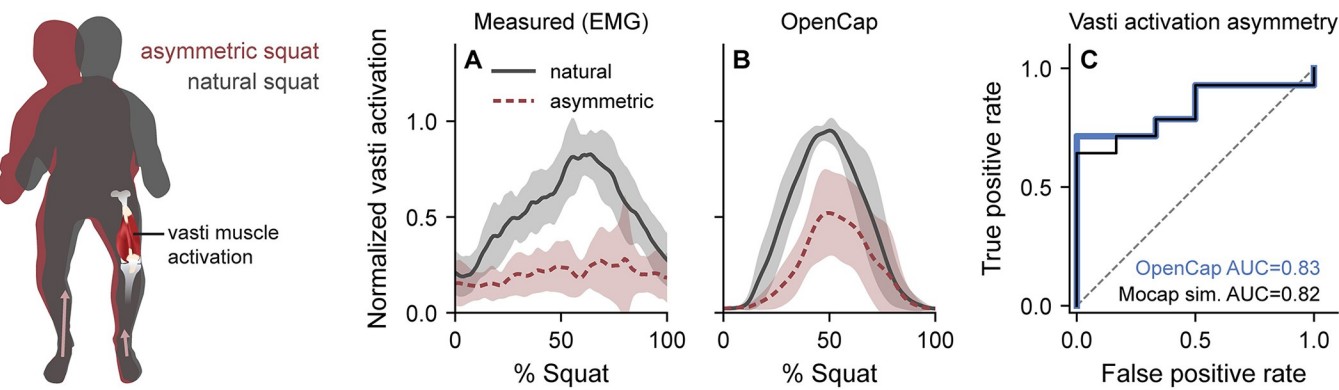

**Fig 5. Asymmetry in vasti muscle activation during squatting.** To assess the utility of OpenCap for informing rehabilitation decisions, we sought to identify between-limb asymmetries in knee extensor muscle (vasti) function that indicate incomplete rehabilitation and relate to poor post-surgical functional outcomes. Participants (n = 10) performed squats naturally, then asymmetrically, where they were instructed to reduce the force under the left foot. **(A, B)** The mean (line) and standard deviation (shading) across participants are shown for the vasti muscle activation of the left (unweighted) leg measured with electromyography (EMG) and estimated using OpenCap. Muscle activations are normalized by the maximum value for each participant and measurement modality. **(C)** OpenCap identified peak vasti activation asymmetry between the left and right leg (asymmetry defined from EMG and clinically relevant symmetry threshold), with area under the receiver operator characteristic curve (AUC) of 0.83 and accuracy of 75%. This was similar to the performance of simulations that used marker-based motion capture and force plate data as input (Mocap sim., AUC = 0.82, accuracy = 70%).

(Fig 5). Since muscle activation can be measured more directly than muscle force, we compared vasti (knee extensor) muscle activation measured with electromyography to activation estimated with OpenCap. We defined ground truth activation asymmetry using electromyography and a clinically relevant symmetry index threshold [43] of 1.15. OpenCap classified squats as being symmetric or asymmetric with an area under the receiver operator characteristic curve (AUC) of 0.83 and an accuracy of 75% at the optimal symmetry index threshold of 1.13 (Fig 5), which was similar to the performance of simulations that used motion capture and force plate data (AUC = 0.82, accuracy = 70%).

## Out-of-lab dynamic analysis

To demonstrate OpenCap's utility in real-world conditions, we extended this analysis of rehabilitation tracking to a field study. A clinician, who was not an expert in movement analysis, used OpenCap to evaluate knee extension moment symmetry in 100 individuals performing natural and asymmetric squats in the community. On average, set up and data collection took five minutes per participant, and for a single squat, kinematics and kinetics were computed automatically in two and 35 minutes on a single server, respectively. In total, data collection took eight hours for 100 subjects, and computation took 31 hours on a 32-thread CPU (kinetic computation was parallelized). OpenCap's peak knee extension moment estimates could discriminate between the symmetric and asymmetric conditions with AUC = 0.90 and accuracy = 85% at the optimal symmetry index threshold of 1.33 when using the condition instruction (i.e., natural or asymmetric) as ground truth (Fig 6A and 6B). OpenCap also detected within-subject improvements in peak knee extension moment symmetry from the asymmetric to the natural condition with AUC = 0.93 and accuracy = 89% at the optimal threshold of 0.26 (Fig 6C and 6D). Together, our lab and field studies demonstrate that OpenCap can detect asymmetries in vasti force generation that may be useful for guiding rehabilitation decisions and can track improvements in symmetry expected to occur over the course of rehabilitation.

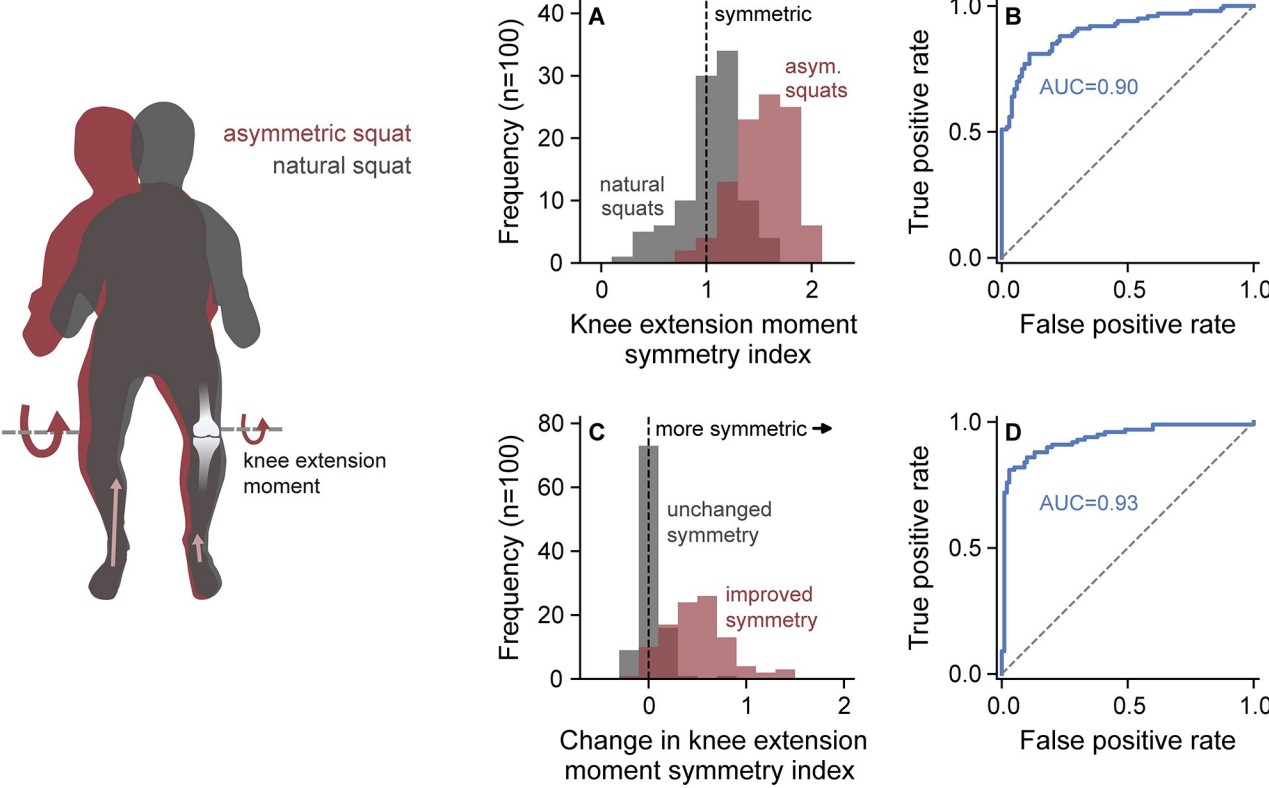

**Fig 6. Out-of-lab dynamic analysis.** To demonstrate the practical utility of OpenCap for tracking rehabilitation progress, we enrolled 100 participants in a clinician-led field experiment. Participants performed symmetric squats and asymmetric squats, where they were instructed to reduce the force under the left foot, which likely resulted in an asymmetry between the left and right knee extension moments. We first evaluated the utility of OpenCap as a screening tool to detect peak knee extension moment asymmetries. **(A)** The distributions of knee extension moment symmetry indices for both squat conditions are shown, with a symmetry index larger than one indicating a lower peak knee extension moment for the left (unweighted) leg compared to the right leg. **(B)** OpenCap's symmetry index estimates classified between natural and asymmetric squats with an area under the receiver operator characteristic curve (AUC) of 0.90 and accuracy of 85%. We then evaluated the utility of OpenCap for detecting changes in peak knee extension moment symmetry that would be expected to occur over time during rehabilitation. **(C)** The distributions of the average difference in the symmetry index between the asymmetric and natural conditions (i.e., hypothetical improved symmetry over time; red) and the average difference in the symmetry index between the three trials in the asymmetric condition (i.e., hypothetical unchanged symmetry over time; gray) are shown. **(D)** OpenCap detected improvements in symmetry from the asymmetric to the natural condition with AUC = 0.93 (improved compared to unchanged distributions from **(C)** and accuracy = 89%.

## Discussion

This study describes OpenCap, a platform that combines computer vision and musculoskeletal simulation to quantify human movement dynamics from smartphone videos. We showed that OpenCap is sufficiently accurate for several research and clinical applications. OpenCap estimated changes in dynamic measures between conditions with similar statistical power (0.65–0.92) as the gold standard technique (0.77–0.89), it estimated dynamic measures that predict adverse outcomes related to osteoarthritis and fall risk with $r^2 = 0.65$–0.80, and it estimated dynamic measures that can inform rehabilitation decision making with classification accuracies of 75–89%. Our field study demonstrated how OpenCap enables clinicians and researchers alike to analyze movement dynamics in the field and in large populations.

OpenCap reduces the cost, time, and expertise barriers to analyzing movement dynamics. OpenCap's hardware can be acquired for between $40 and $700, depending on whether users need to purchase new iOS devices (Fig 1). This is about 215 times cheaper than traditional motion capture laboratory equipment, and it does not require a dedicated laboratory space.

Hands-on time for measuring movement dynamics with OpenCap is the time for setting up the mobile devices and the time to perform the movements (Fig 1, bottom row). During our in-field data collection, the five minutes of hands-on time per participant was about 25 times less than a comparable analysis in a motion capture laboratory (about 2 hours). OpenCap does not require specialized software or expertise, which bridges gaps between the computer vision, biomechanics, and clinical movement science communities. Most computer vision algorithms require computer science knowledge to run and most simulation tools require biomechanics knowledge to operate, but OpenCap automates these processes, making advancements in these fields more accessible to clinicians and researchers. In the year since releasing OpenCap, over 2,000 researchers from a variety of fields have used it to collect tens of thousands of motion trials. OpenCap also meets Stanford University's security requirements for cloud-based systems using high-risk data (e.g., protected health information) and ensures end-to-end data encryption. To further facilitate ease of use, we provide tutorials and examples on a companion website (https://opencap.ai/).

Our results demonstrate OpenCap's potential clinical utility as a screening tool and for informing rehabilitation decisions. Future studies could test OpenCap's ability to screen for risk of non-contact ligament injury in athletes [11,44], or to predict efficacy of surgery in individuals with cerebral palsy [12,13]. OpenCap assessments may also be fast enough to enable movement screens to become part of routine clinical care, allowing clinicians to track function over time, and following an injury or surgery, to benchmark rehabilitation status against pre-injury measures [45]. We investigated OpenCap's accuracy for estimating several clinically meaningful metrics; we encourage further validation studies for other activities, populations, and metrics of interest to evaluate whether OpenCap is sufficiently accurate to inform decisions in use cases beyond those that we explored.

By enabling large-scale, out-of-lab studies, OpenCap can accelerate movement research. OpenCap detected between-condition differences with similar statistical power as motion capture and force plate analysis, but in substantially less time. This accuracy and efficiency makes prospective injury risk studies that require hundreds of participants more feasible, enables the incorporation of movement dynamics into population-scale health studies that typically only use pedometry (e.g., the Osteoarthritis Initiative [46] or the UK Biobank [47]), and facilitates the development of more sensitive functional outcome measures for clinical trials. By automatically computing kinematics, OpenCap is not susceptible to errors introduced by between-experimenter variance in motion capture marker placement [48]. This could reduce variability in multi-center studies [49] and enable movement data to be compiled into homogeneous, sharable datasets that are useful to the machine learning community. Importantly, OpenCap's portability will enable studies of populations that are often underrepresented in movement research due to time and geographic constraints.

OpenCap uses machine learning to improve the fidelity of video-based kinematic estimates and physics-based modeling to maintain generalizability (Fig 2). We combined a deep learning model suited for time series data with a constrained biomechanical model to estimate 3D kinematics from video keypoints that, alone, are insufficient to characterize 3D kinematics. Our deep learning model predicts a comprehensive set of anatomical markers from the sparse video keypoints labeled in common computer vision datasets (e.g., COCO [50]). Using the predicted anatomical markers, instead of video keypoints, with our biomechanical model improved kinematic accuracy, averaged across all degrees of freedom, by 3.4˚ (see Methods and S2 Table). These improvements were greatest for the hip flexion, pelvic tilt, and lumbar flexion degrees of freedom (4.9–32.6˚ improvement), which are susceptible to large angular errors due to the sparsity of video keypoints between the hips and shoulders. Additionally, the deep learning model architecture provides temporal consistency, making OpenCap more

robust to brief occlusions or mis-identified keypoints. Finally, despite some task-dependent tuning of the problem formulation, muscle-driven dynamic simulations are a more generalizable approach for estimating dynamics than an end-to-end machine learning approach. This enabled us to study the dynamics of different movements without training data for each activity.

The accuracy of OpenCap's kinematic and kinetic estimates is similar to state-of-the art markerless motion capture solutions. OpenCap's kinematic error (range of root mean squared error [RMSE] across lower-extremity degrees of freedom: 2.0–10.2˚) is similar to errors reported for inertial measurement unit–based approaches (RMSE: 2.0–12˚ for walking, running, and daily living activities [18,51–57]) and commercial and academic video-based systems with eight cameras (RMSE: 2.6–11˚ for walking, running, and cycling activities [31,58]). Additionally, the joint angle errors between OpenCap and marker-based motion capture are similar to errors induced by skin motion artifact in marker-based motion capture. For example, the knee flexion angle computed with skin-mounted markers can have a root-mean-square error of up to 8˚ compared to the angle measured with bone pin–mounted markers [59], which exceeds the 4.2˚ root-mean-square error in knee flexion angle between OpenCap and skin marker–based motion capture (S2 Table). Dramatic improvements in the concordance between markerless and marker-based motion capture systems may be challenging to demonstrate due to the errors in the skin marker–based motion capture approach used as ground truth. Furthermore, in contrast with most inertial measurement unit–based approaches, OpenCap estimates global translations (e.g., pelvis displacement), enabling estimation of whole-body measures like center-of-mass trajectory. Interestingly, kinematic estimates did not substantially improve when using more than two cameras (see Methods and S2 Table), suggesting that two cameras are sufficient for analyzing activities like those included in this study. To our knowledge, there is no previous example of computing whole-body kinetics from video alone; however, OpenCap's kinetic estimates are comparable to inertial measurement unit–based approaches. For example, OpenCap's root mean square errors in ground reaction force (1.5–11.1% bodyweight) and lower-extremity joint moment (0.3–1.7% bodyweight*height) predictions during walking (S3 and S4 Tables) are comparable to those resulting from using a 17-sensor inertial measurement unit suit (1.7–9.3% bodyweight and 0.5–2.2% bodyweight*height, respectively [18,60]). OpenCap also predicted the first peak knee adduction moment during walking with 44% higher accuracy than a machine learning model trained specifically to predict this measure from marker positions that could be extracted from video [24].

OpenCap transforms the outputs of pose detection algorithms into valuable insights for studying human movement. We designed OpenCap to integrate different pose detection algorithms, and we found only minor differences in kinematics when testing different algorithms (see Methods and S1 and S2 Tables). With the recent advances in joint center estimation from single-view [26,30,61,62] and multi-view [26–30] video, we expect OpenCap's accuracy in estimating kinematics and kinetics to improve as more accurate pose estimation algorithms are released. By sharing our data and source code, we encourage researchers to benchmark their models using our data and to contribute to OpenCap's development by adding support for their models.

Our study has several limitations. First, we tested OpenCap's ability to estimate informative dynamic measures by having healthy individuals simulate different movement patterns associated with pathology or treatment. While the simulated movements were similar to those reported in the populations of interest (see Methods: Applications and Statistics), and OpenCap could distinguish differences in dynamics between these simulated conditions, future work is needed to validate these measures in the populations of interest. Furthermore, the relationship between some kinetic measures and clinical outcomes has been established in the

literature; however, the level of accuracy at which these metrics need to be estimated to inform clinical decisions is unknown in most cases because establishing these thresholds requires resource-intensive longitudinal studies. Where possible, we compared OpenCap's accuracy to clinically meaningful thresholds (e.g., the knee adduction moment). We hope that OpenCap's ease of use will enable more longitudinal studies that can establish the clinical utility of kinetic metrics and the required measurement accuracy. Second, we did not measure the repeatability of measurements obtained with OpenCap. However, based on previous repeatability studies using markerless motion capture and employing similar methodologies [49], we anticipate OpenCap to measure kinematics with similar repeatability to, if not better than, marker-based motion capture. Future studies are necessary to confirm this claim and assess repeatability for dynamics assessments. Third, the deep learning model that augments our 3D marker set may not generalize to activities outside of the distribution of activities that it was trained on. We generated the training data for this model using standard OpenSim kinematics data, so additional datasets could be added to the training set in the future. Additionally, estimating dynamics requires some task-dependent user inputs, which is a limitation of any optimization-based muscle-driven simulation. We have provided optimization problem formulations that work well for several activities. Overall, if future applications require high accuracy rather than generalizability, OpenCap's accuracy could likely be improved with task-specific tuning of the deep learning model and optimization problem formulation.

In conclusion, OpenCap allows non-experts to analyze human movement dynamics in an order of magnitude less time and for several orders of magnitude less money than was previously possible with marker-based motion capture and force plates. We expect that OpenCap will catalyze large-scale studies of human movement, the sharing of motion datasets, and the translation of movement biomarkers into clinical practice.

## Methods

### Ethics statement

The protocol for this study was approved and overseen by the Institutional Review Board of Stanford University (IRB00000351). We conducted the experiment in accordance with this approved protocol and relevant guidelines and regulations. All participants provided written informed consent before participation. Participants whose identifiable videos are shared publicly consented to sharing these videos.

### Design

OpenCap comprises several steps to estimate movement dynamics from videos. These steps include calibrating cameras, collecting and processing videos, estimating marker positions, estimating kinematics, and generating physics-based dynamic simulations of movements. This pipeline is implemented in Python (v3.7.10). OpenCap's web application guides users through each step, and cloud instances are used for computing (Fig 2).

**Camera calibration.** OpenCap models the iOS device cameras using a fifteen-parameter pinhole camera model (https://github.com/smidm/camera.py) and computes parameters using OpenCV [63]. At the beginning of a data collection, OpenCap loads the pre-computed intrinsic parameters related to each device's camera hardware and recording settings (principal point, focal length, and distortion parameters) from a database that we created of recent iOS devices. Next, the web application guides users to place a checkerboard in view of all cameras, and OpenCap automatically computes the extrinsic parameters (camera transformation relative to the global frame) from a single image of a checkerboard. We used a precision-manufactured 720x540 mm checkerboard to pre-compute the intrinsic camera parameters for each

device in our database (see S1 Appendix for details about intra- and inter-phone intrinsic parameter testing). A 210x175 mm checkerboard printed on A4 paper and mounted to a flat surface is sufficient for computing extrinsic camera parameters during each data collection. We found minimal kinematic differences when using the printed checkerboard, compared to the precision-manufactured checkerboard, to calibrate the cameras (see S1 Appendix).

**Video collection and pose estimation.**    After calibration, users can proceed with simultaneously recording videos on all devices through the web application. Videos are recorded at a resolution of 720x1280 pixels, a frame rate of 60 Hz, and with the camera focus distance set to a fixed value.

Recorded videos are then processed using video pose detection algorithms. OpenCap currently supports two algorithms: OpenPose [21] and HRNet [64–67]. These algorithms were selected due to performance and the inclusion of foot keypoints. For each video, and at each time frame, both algorithms return the two-dimensional (2D) position of body keypoints as well as a confidence score (between 0 and 1) indicating the confidence of the algorithm in the keypoint position. Twenty body keypoints are included for further analysis (neck, mid hip, left and right shoulders, hips, knees, ankles, heels, small and big toes, elbows, and wrists). OpenCap implements custom algorithms for processing 2D keypoint positions (e.g., handling keypoint occlusion) and time synchronizing them across videos using cross-correlations of keypoint velocities (see S1 Appendix for details).

**Triangulation and marker-set augmentation.**    OpenCap triangulates the synchronized 2D video keypoint positions to compute 3D positions. OpenCap uses a Direct Linear Transformation algorithm for triangulation [68], and weights the contribution of individual cameras in the least-squares problem with the corresponding keypoint confidence score [58]. There are two major limitations of using 3D keypoint positions triangulated from video for biomechanical analysis. First, the video keypoint set is not sufficient to fully define the kinematics of all degrees-of-freedom of the body segments. Tracking these limited keypoints using a model with biomechanical joint constraints mitigates this issue for some, but not all body segments. For example, keypoints at the hips and shoulders are insufficient for robustly determining sagittal-plane hip, pelvis, and lumbar kinematics. Second, most pose estimation algorithms identify keypoints on a frame-by-frame basis, so the resulting 3D keypoint trajectories are often physically unrealistic, especially in the presence of misidentified or occluded keypoints.

To overcome these limitations, we trained two long short-term memory (LSTM) networks to predict the 3D positions of 43 anatomical markers from the 3D positions of the 20 triangulated video keypoints. The set of anatomical markers corresponds to what is commonly used for marker-based motion capture [69] to robustly determine 3D joint kinematics. We chose LSTM networks as they leverage time series data, which may improve the temporal consistency of the output marker position trajectories. We trained two LSTM networks: an *arm model* to predict the positions of eight arm markers from the positions of seven arm and torso keypoints, and a *body model* to predict the positions of 35 body markers from the positions of 15 lower-limb and torso keypoints. Both models also use height and weight as inputs. To train the networks, we synthesized corresponding pairs of 3D video keypoints and 3D anatomical markers from 108 hours motion capture data processed in OpenSim from published biomechanics studies [70–79] (see S1 Appendix for details on dataset generation). We split the data into a training set (~80%), validation set (~10%), and test set (~10%). Prior to training, we expressed the 3D positions of each marker with respect to a root marker (the midpoint of the hip keypoints), normalized the 3D positions by the subject's height, sampled at 60 Hz, split the data into non-overlapping time-sequences of 0.5 s, and added Gaussian noise (standard deviation: 18 mm) to each time step of the video keypoint positions based on a range of previously reported keypoint errors [24,26,29]. For both models, we tuned hyperparameters using a

random search. The RMSEs on the test set were 8.0 and 15.2 mm for the body and arm model, respectively (see S1 Appendix for details about model architecture and training). In practice, OpenCap uses both LSTM networks to predict root-centered arm and body anatomical marker positions from root-centered 3D video keypoints. It then adds the root keypoint position to all predicted positions.

**Physics-based modeling and simulation.** After calibration, OpenCap's web application guides users to record the participant in a standing neutral pose. OpenCap uses the anatomical marker positions estimated from the neutral pose to scale a musculoskeletal model to the participant's anthropometry using OpenSim's Scale tool. OpenCap uses the musculoskeletal model from Lai et al. [69,80] with modified hip abductor muscle paths according to Uhlrich et al. [79]. The musculoskeletal model comprises 33 degrees of freedom (pelvis in the ground frame [6], hips [2x3], knees [2x1], ankles [2x2], metatarsophalangeal joints [2x1], lumbar [3], shoulders [2x3], and elbows [2x2]). Note that since no markers are attached to the toes, no reliable estimates of metatarsophalangeal joint kinematics can be obtained. The metatarsophalangeal joint is nevertheless included when generating tracking simulations, since modeling that joint improves knee mechanics in muscle-driven simulations [81]. The musculoskeletal model is driven by 80 muscles actuating the lower-limb coordinates and 13 ideal torque motors actuating the lumbar, shoulder, and elbow coordinates. Ground reaction forces (i.e., external forces) are modeled through six foot-ground contact spheres attached to the foot segments of the model [82,83]. Raasch's model [84,85] is used to describe muscle excitation-activation coupling, and a Hill-type muscle model [86,87] is used to describe muscle-tendon dynamics and the dependence of muscle force on muscle fiber length and velocity. Skeletal motion is modeled with Newtonian rigid body dynamics and smooth approximations of compliant Hunt-Crossley foot-ground contacts [88,89]. The dynamics of the ideal torque motors are described using linear first-order approximations of a time delay [83]. To increase computational speed, muscle-tendon lengths and velocities, and moment arms are defined as a polynomial function of joint positions and velocities [90]. The polynomial coefficients are fit to the output from OpenSim's Muscle Analysis tool applied to 5000 randomly varied lower limb postures. Muscles are represented by ninth-order or lower polynomials, with RMSE of muscle-tendon length and moment arm lower than 1.5 mm compared to the original model.

After scaling, users can record any movement through OpenCap's web application. OpenCap then uses the anatomical marker positions estimated from the recorded videos and LSTM network to compute joint kinematics using OpenSim's Inverse Kinematics tool and the scaled musculoskeletal model. Users can visualize the resulting 3D kinematics in the web application.

Finally, OpenCap can estimate dynamics using muscle-driven tracking simulations of joint kinematics. The tracking simulations are formulated as optimal control problems that aim to identify muscle excitations that minimize a cost function subject to constraints describing muscle and skeleton dynamics. The cost function $J$ (Eq 1) includes squared terms for muscle activations ($a$) and excitations of the ideal torque motors at the lumbar, shoulder, and elbow joints ($e_{tm}$). It also includes tracking terms (squared difference between simulated and reference data), namely tracking of experimental joint positions ($\tilde{q}$), joint velocities ($\dot{\tilde{q}}$), and joint accelerations ($\ddot{\tilde{q}}$):

$$J = \int_{t_0}^{t_f} w_1 a^2 + w_2 e_{tm}^2 + w_3 \|\tilde{q} - q\|_2^2 + w_4 \|\dot{\tilde{q}} - \dot{q}\|_2^2 + w_5 \|\ddot{\tilde{q}} - \ddot{q}\|_2^2 dt, \tag{1}$$

where $t_0$ and $t_f$ are initial and final times, $w_i$ with $i = 1,\ldots,5$ are weights, and $t$ is time. Experimental joint positions, velocities, and accelerations are low-pass filtered using fourth-order, zero-lag Butterworth filters (default cutoff frequencies are 12 Hz for gait trials and 30 Hz for

non-gait trials). Each cost term is scaled with empirically determined weights. To avoid singular arcs [91], a penalty function is appended to the cost function with the remaining control variables [83,92]. Note that the optimal control problem formulation can be tailored to the activity of interest to incorporate activity-based knowledge by, for instance, adjusting the cost function, constraints, and filter settings (see S1 Appendix). The optimal control problems are formulated in Python with CasADi [93] (v3.5), using direct collocation and implicit formulations of the muscle and skeleton dynamics [83]. Algorithmic differentiation is used to compute derivatives [92], and IPOPT is used to solve the resulting nonlinear programming problems [94] with a convergence tolerance of 1e-4 (all other settings are kept to default).

**Practical considerations.** The outcomes of OpenCap can be influenced by environmental and experimental factors. To maximize the accuracy of results, we recommend users follow several best practices. First, the pose estimation models perform best when the participant is clearly visible in the video. Enhancing visibility can be achieved by wearing clothing articles that contrast with each other and the background. For example, blue pants and white shoes on a black floor would yield better results than black pants and black shoes on a black floor. It is also advisable for the participant to wear tight-fitting clothing to facilitate the detection of their joints. While these considerations likely improve OpenCap's performance, to best represent out-of-lab data-collection conditions, we did not ask participants in this study to wear specific colors of clothing. Participant visibility can also be enhanced by avoiding excessive brightness within the field of view and choosing an uncluttered background.

Second, OpenCap performs best when all body segments are visible by at least two cameras at all times. OpenCap can handle temporary occlusions of body segments, but it is advisable to minimize occlusions by optimizing the position of the cameras. When using two cameras for activities with minimal subject displacement (e.g., squats, sit-to-stands, and drop jumps), we recommend positioning the cameras at approximately ±45˚ from the subject's forward-facing direction. For activities like walking, placing the two cameras at about ±30˚ from the subject's walking direction can reduce occlusions and increase the length of the capture volume.

Third, when using two cameras, the size of the capture volume is maximized when the subject enters the field of view of both cameras at the same location in the volume and exits the field of view of both cameras at the same location. For example, when positioning cameras for walking, the participant can stand where the capture volume should begin, and both cameras can be adjusted such that the participant would just enter the field of view of both cameras at this location.

Fourth, the distance between the subject and the cameras affects the accuracy of pose estimation models. We recommend selecting a distance such that the subject's joints are clearly discernible in the recorded videos. Performing movements 2–10m from the cameras works well; however, higher resolution pose estimation settings may be needed for accurate kinematics at 10m (i.e., HRNet or OpenPose high-accuracy settings). Furthermore, extrinsic calibration errors increase as distance increases from the calibration board's position. Performing activities near where the board was placed during calibration or using a larger calibration board can mitigate these problems. We also recommend not positioning the cameras less than 2m from the subject, since some pose estimation models commonly fail when the subject takes up a significant portion of the field of view.

Finally, we recommend users frequently visit our website opencap.ai, where we provide best practices and updates about new features. In the year following its release, we have added multiple features to OpenCap including support for different musculoskeletal models, recording videos at different frame rates and in landscape mode, and interactive data plotting.

## Validation

**Participants and experiment.**    To validate OpenCap against gold standard kinematic and kinetic measures, we measured ten healthy adults (sex: 6 female and 4 male; age = 27.7±3.8 [23-25] years; body mass = 69.2±11.6 [59.0–92.9] kg; height = 1.74±0.12 [1.60–1.96] m; mean ± standard deviation [range]) performing multiple activities in a motion capture laboratory. Participants were instructed to perform four activities in a natural (i.e., self-selected) and modified way during data collection: i) walking naturally and with a trunk sway modification (trunk leaned laterally over stance leg), ii) performing five squats naturally and then asymmetrically (reduced force under the left foot), iii) performing five sit-to-stands naturally and then with increased trunk flexion (forward lean when rising), and iv) performing three drop jumps naturally and then asymmetrically (reduced force under the left foot when landing).

**Experimental data.**    We measured ground truth kinematics, ground reaction forces, and muscle activity with optical motion capture, force plates, and electromyography. An eight-camera motion capture system (Motion Analysis Corp., Santa Rosa, CA, USA) tracked the positions (100 Hz) of 31 retroreflective markers placed bilaterally on the 2nd and 5th metatarsal heads, calcanei, medial and lateral malleoli, medial and lateral femoral epicondyles, anterior and posterior superior iliac spines, sternoclavicular joints, acromia, medial and lateral epicondyles of the humerus, radial and ulnar styloid processes, and the C7 vertebrae. Twenty additional markers were used to aid in segment tracking. Ground reaction forces were synchronously measured (2000 Hz) using three in-ground force plates (Bertec Corp., Columbus, OH, USA). Wireless electromyography electrodes (Delsys Corp., Natick, MA, USA) measured muscle activity (2000 Hz) from the vastus lateralis and medialis (electromyography data from 14 other lower-extremity muscles are shared with the dataset but not analyzed here). We used OpenCap to record video from five smartphones (iPhone 12 Pro, Apple Inc., Cupertino, CA, USA). The phones were positioned 1.5 m off the ground, 3 m from the center of the force plates, and at ±70˚, ±45˚, and 0˚, where 0˚ faces the participant. Unless otherwise noted, the validation results used only the two ±45˚ cameras. A precision-manufactured, 720x540 mm checkerboard was used for computing the extrinsic parameters during OpenCap's camera calibration step.

Marker, force, and electromyography data were filtered using a fourth-order, zero-lag Butterworth filter. Marker and force data were low-pass filtered (walking: 6 Hz, squat: 4 Hz, sit-to-stand: 4 Hz, and drop jump: 30 Hz). These frequencies were selected as the frequency that retained 99.7% of the cumulative signal power of the Fourier-transformed marker trajectories [95]. Electromyography data were band-pass filtered (30–500 Hz), rectified, and low-pass filtered (6 Hz). Electromyography data were normalized to maximum activation trials including maximum height jumps, sprinting, and isometric and isokinetic ankle dorsiflexion, knee flexion, hip abduction exercises [96].

**Kinematics and kinetics.**    Laboratory-based (later referred to as Mocap) kinematic and kinetic data were estimated from measured marker and force plate data using OpenSim 4.3. We used the same modeling and simulations pipeline as OpenCap to scale the musculoskeletal models and estimate joint kinematics from measured marker data (see Methods: Design: Physics-based modeling and simulation). Joint kinetics were then estimated from joint kinematics (filtered at same frequencies as force plate data) and force plate data using OpenSim's Inverse Dynamics tool.

OpenCap kinematic and kinetic data were estimated using the two 45˚ cameras and the HRNet pose detection algorithm. This setup combines simplicity, performance, and a permissible open-source software license. It was selected after conducting a sensitivity analysis studying the effect of using different camera configurations (two, three, and five cameras) and pose

detection algorithms (OpenPose with default settings, OpenPose with high-accuracy settings, and HRNet) on predicted anatomical marker positions and joint kinematics. See S1 Appendix and Methods: Validation: Validation Results for details about the sensitivity analysis and pose detection algorithm settings.

**Error analysis.**   We evaluated the performance of OpenCap against Mocap by quantifying errors in anatomical marker positions, joint kinematics, ground reaction forces, and joint kinetics.

We quantified errors in 3D anatomical marker positions using mean per marker error (Euclidean distance). We report errors for 17 anatomical markers (the C7 vertebrae and the left and right acromia, anterior and posterior superior iliac spines, medial and lateral femoral epicondyles, medial and lateral malleoli, calcanei, and second and fifth metatarsal heads). Prior to error analysis, we synchronized and aligned Mocap and OpenCap position data by removing the time delay that minimized the mean difference between marker positions (averaged over all markers and time steps), then subtracting this average position offset from the OpenCap positions.

We quantified errors in 3D joint kinematics using MAE. We report errors for 18 rotational degrees of freedom (pelvis rotations [3], hips [2x3], knees [2x1], ankles [2x2], and lumbar [3]) and three translational degrees of freedom (pelvis translations).

We quantified errors in 3D ground reaction forces using MAE normalized by bodyweight. We also expressed errors as percent of range of the measured signal over each trial. Prior to quantifying errors, we filtered ground reaction forces from OpenCap using the same filters as for the measured ground reaction forces (see Methods: Validation: Experimental data).

We quantified errors in 3D joint kinetics using MAE normalized by bodyweight times body height. We report errors for 15 rotational degrees of freedom (hips [2x3], knees [2x1], ankle [2x2], and lumbar [3]). It is important to note that while joint moments estimated from inverse dynamics are considered gold standard, they include non-physical pelvis residual forces and moments to compensate for the inconsistency between model-based kinematics and measured ground reaction forces. In contrast, muscle-driven simulations are dynamically consistent and do not include pelvis residuals. Thus, the differences between Inverse Dynamics and OpenCap-estimated joint moments are not entirely attributable to error in the OpenCap pipeline.

**Validation results.**   The marker error, averaged across markers and activities, was 32 mm using the two-camera HRNet setup. Our sensitivity analysis demonstrated that OpenCap's accuracy remained consistent across different pose detectors and additional cameras. Marker error was 31 and 35 mm when using OpenPose with high-accuracy settings and default setting, respectively. Using three cameras did not improve accuracy, but using five cameras mildly reduced error (29 mm for HRNet). Marker error was larger for the upper extremity (39 mm) and pelvis (38 mm) than for the lower extremity (27 mm) using the two-camera HRNet setup. Detailed results of the sensitivity analyses are presented in S1 Table.

The kinematic MAE for the two-camera HRNet setup, averaged across degrees of freedom and activities, was 4.5˚ (range = 1.7–10.3˚) and 12.3 mm (range = 5–20.3 mm) for the 18 rotational and three translational degrees of freedom, respectively (Table 1). Our sensitivity analysis showed that kinematic errors were similar when using the high-accuracy settings and default OpenPose settings (4.3˚ and 4.7˚, respectively), and when adding cameras (improvement of less than 0.3˚). We also investigated the effect of using video keypoints instead of anatomical markers to estimate joint kinematics. Kinematic errors were 3.4˚ worse on average for the two-camera HRNet setup when using the video keypoints instead of the anatomical markers. This was primarily due to 12.1–39.2˚ errors at the lumbar extension, pelvic tilt, and hip flexion degrees of freedom, due to the limited information in the video keypoint marker set for

distinguishing between rotations at these joints. Detailed results of the sensitivity analyses are presented in S2 Table. Average kinematic waveforms estimated using OpenCap and Mocap are presented in Figs A–D in S1 Appendix.

The ground reaction force MAE, averaged across directions and activities was 6.2% bodyweight. It was 11.4% bodyweight in the vertical direction, 3.5% bodyweight in the anterior-posterior direction, and 3.8% bodyweight in the medio-lateral direction (Table 1). The joint moment MAE, averaged across degrees of freedom and activities, was 1.2% bodyweight*height (Table 1). Detailed results are presented in S3 and S4 Tables, and average ground reaction force and joint moment waveforms estimated using OpenCap and Mocap are presented in Figs E–L of S1 Appendix.

## Applications and statistics

We assessed OpenCap's ability to estimate kinetic measures related to musculoskeletal pathology in three applications that represent clinical use cases. Unless otherwise noted, these analyses were performed on the 10-subject dataset described in Methods: Validation. All statistical analyses were performed in Python (v3.7.10) using the scipy [97] (v1.5.4), statsmodels [98] (v0.13.2), and pingouin [99] (v0.5.2) packages. We compared conditions within and between measurement modalities using $r^2$, MAE, two-sided paired t tests (alpha = .05), and two-sided Wilcoxon signed rank tests. Prior to conducting a t test, we tested for normality using a Shapiro Wilkes test, and we used a Wilcoxon signed rank test to compare non-normally distributed data. To prevent inflated Type 1 error from multiple comparisons, we report corrected P-values after controlling for the false discovery rate using the Benjamini Hochberg procedure [100]. We evaluated the post-hoc power of t tests and Wilcoxon signed rank tests using the sample size, alpha = .05, and the observed effect size. We evaluated performance on classification tasks using AUC and binary classification accuracy at the threshold that maximized the true positive rate minus the false positive rate. Unless otherwise noted, values are reported as mean ± standard deviation.

In the first application, we assessed the peak knee adduction moment and peak medial knee contact force during walking. Participants walked naturally and with a trunk sway modification, which typically alters medial knee loading [32]. Participants walked with 15˚ more trunk sway on average during the trunk sway compared to the natural condition, which is similar to the 10–13˚ of trunk sway reported in gait modification studies [32,101]. We computed peaks of both loading measures during the first half of the stance phase using the Joint Reaction Analysis tool in OpenSim (see S1 Appendix for details), which uses kinematics, ground reaction forces, and muscle forces as inputs. For OpenCap, we used the outputs of the muscle-driven dynamic simulation for this analysis, and for Mocap, we used the OpenSim Static Optimization tool to estimate muscle forces. We used static optimization for the Mocap data as it is commonly used to estimate knee contact forces during low-frequency tasks like walking, and it is sensitive to changes in contact force induced by gait modifications [102]. We first determined how accurately OpenCap could estimate the peak knee adduction moment and how it varies among gait patterns and individuals. For each walking condition, we averaged the peak knee adduction moment across the three trials for each individual and compared between OpenCap and Mocap using $r^2$ and MAE. We then determined whether OpenCap could detect group changes in both loading measures from a gait modification similarly to Mocap. For each measurement modality, we used either a two-sided paired t test or a Wilcoxon signed rank test to evaluate the changes from baseline, and we computed the post-hoc power of each test. Finally, we evaluated whether OpenCap correctly identified an increase or decrease in peak knee loading measures for each individual, using the Mocap estimate as ground truth.

In the second application, we evaluated lower-extremity joint moments while rising from a 40 cm chair. Participants stood naturally and with increased trunk flexion, which can shift the muscle force demand from the knee extensors to the hip extensors and ankle plantarflexors [40]. During the increased trunk flexion condition, participants stood with a 42±8˚ of trunk flexion, which is similar to the 47˚ reported in a cohort of older adults with functional limitations [103]. For three repetitions per condition, we averaged the hip extension, knee extension, and ankle plantarflexion moments over the rising phase, then averaged these values across repetitions. To evaluate OpenCap's ability to detect group changes between conditions, we compared the moment changes from baseline to trunk-lean using two-sided paired t tests for both OpenCap and Mocap. We then conducted a post-hoc power analysis for each measurement modality. To determine OpenCap's ability to identify individuals with low knee extensor moments during this motion, we compared each participant's average knee extension moment for each condition between OpenCap and Mocap using $r^2$ and MAE.

In the third application, we assessed the between-limb symmetry of knee extensor muscle activation while squatting. Participants squatted naturally and asymmetrically, which can elicit asymmetrical knee extensor force generation [7]. We first performed an in-lab experiment to compare peak vasti muscle (knee extensors) activation measured with electromyography to peak activation estimated with OpenCap and Mocap. Since there is no change in muscle strength between these conditions, a change in muscle activation between conditions is a more easily measured surrogate for a change in muscle force. For OpenCap, muscle activations were outputs of the muscle-driven tracking simulations, whereas for Mocap, muscle activations were estimated using OpenSim's Static Optimization tool. We first averaged the activation of the vastus medialis and vastus lateralis, then extracted the peak value over a squat (standing to standing again). We calculated the peak vasti activation symmetry index between the left and right leg (Eq 2) and averaged across three repetitions in each condition:

$$symmetry\ index = 1 - \frac{a_{\text{involved}} - a_{\text{uninvolved}}}{a_{\text{uninvolved}}}, \tag{2}$$

where $a_{\text{involved}}$ is the peak activation of the left vasti (reduced force under left foot during asymmetric condition) and $a_{\text{uninvolved}}$ is the peak activation of the right vasti. The symmetry index is larger than one when the left peak vasti activation is lower than the right peak vasti activation, which would be expected in the asymmetric condition. On average, during the asymmetric condition compared to the natural condition, our participants squatted with a 0.53±0.32 greater symmetry index measured by electromyography; this is similar to the 0.51 greater asymmetry in vasti strength reported in individuals one month after a total knee replacement [104]. We determined OpenCap's ability to classify symmetric vs. asymmetric squats using AUC and classification accuracy, with ground truth symmetry labels determined from electromyography based on a symmetry index threshold (1.15) that predicts functional deficits following anterior cruciate ligament surgery [43]. We also computed the AUC and accuracy for simulated muscle activations from Mocap.

Finally, we performed a field study where a clinician used OpenCap to evaluate knee extension moment symmetry in 100 individuals outside of the laboratory (sex: 41 female and 59 male; age = 29.6±9.2 [18-67] years, body mass = 69.2±12.0 [50–109] kg; height = 1.74±0.09 [1.45–1.97] m; mean ± standard deviation [range]). We used a 210x175 mm checkerboard printed on A4 paper and mounted to plexiglass for camera calibration. Participants performed natural squats and asymmetric squats. All participants provided written informed consent before participation. The study protocol was approved and overseen by the Institutional Review Board of Stanford University (IRB00000351). We conducted the experiment in accordance with this approved protocol and relevant guidelines and regulations. First, we evaluated OpenCap's ability to detect a squat with a between-limb asymmetry in the peak knee extension moment. For each participant,

we computed the peak knee extension moment for three repetitions per condition, computed the peak knee extension moment symmetry index (Eq 2), and averaged across the repetitions in each condition. To determine the classification performance, we computed AUC and accuracy, with ground truth labels being the instructed condition (i.e., natural [assumed to be symmetric] vs. asymmetric squats). Second, we evaluated OpenCap's ability to detect between-condition changes in knee extension moment symmetry, simulating the ability to detect improvements in symmetry that would be expected to occur over time. To simulate improved symmetry, we subtracted each participant's symmetry index averaged over the repetitions of the natural condition from their symmetry index averaged over repetitions of the asymmetric condition (a positive value indicates an improvement in symmetry). To simulate unchanged symmetry, we averaged the difference in symmetry index between each combination of the asymmetric squat repetitions. We computed the AUC and accuracy of this change in symmetry measure using the known class (i.e., improved symmetry or unchanged symmetry) as ground truth.

## Supporting information

**S1 Appendix. Supplementary methods and results.**
(PDF)

**S1 Table. Errors in each marker position between OpenCap and motion capture.** The mean per-marker error is shown for each marker, activity, camera combination, and pose detection algorithm.
(XLSX)

**S2 Table. Errors in kinematics between OpenCap and motion capture.** The mean absolute error (MAE) and root mean square error (RMSE) are shown for each degree of freedom, activity, camera combination, and pose detection algorithm.
(XLSX)

**S3 Table. Errors in ground reaction forces between OpenCap and force plates.** The mean absolute error (MAE), root mean square error (RMSE), and mean absolute error as a percentage of the range (MAPE) are shown for each activity using the two-camera HRNet setup.
(XLSX)

**S4 Table. Errors in joint moments between OpenCap and inverse dynamics using motion capture and force plates.** The mean absolute error (MAE), root mean square error (RMSE), and mean absolute error as a percentage of the range (MAPE) are shown for each activity and degree of freedom using the two-camera HRNet setup.
(XLSX)

**S1 Movie. Overview of the motivation, technology, and impact of OpenCap.** We introduce the value and limitations of the current lab-based approach for estimating human movement dynamics. We then show an OpenCap data collection and demonstrate how it overcomes the cost, time, and expertise limitations of the lab-based approach. Finally, we demonstrate Open-Cap's accuracy and discuss how it can catalyze large-scale research studies and the clinical translation of movement biomarkers.
(MP4)

## Author Contributions

**Conceptualization:** Scott D. Uhlrich, Antoine Falisse, Łukasz Kidziński, Jennifer L. Hicks, Scott L. Delp.

**Data curation:** Scott D. Uhlrich, Antoine Falisse, Łukasz Kidziński, Julie Muccini.

**Formal analysis:** Scott D. Uhlrich, Antoine Falisse, Łukasz Kidziński.

**Funding acquisition:** Akshay S. Chaudhari, Jennifer L. Hicks, Scott L. Delp.

**Investigation:** Scott D. Uhlrich, Antoine Falisse, Łukasz Kidziński, Julie Muccini.

**Methodology:** Scott D. Uhlrich, Antoine Falisse, Łukasz Kidziński, Jennifer L. Hicks, Scott L. Delp.

**Project administration:** Jennifer L. Hicks, Scott L. Delp.

**Resources:** Jennifer L. Hicks, Scott L. Delp.

**Software:** Scott D. Uhlrich, Antoine Falisse, Łukasz Kidziński, Michael Ko.

**Supervision:** Jennifer L. Hicks, Scott L. Delp.

**Validation:** Scott D. Uhlrich, Antoine Falisse.

**Visualization:** Scott D. Uhlrich, Antoine Falisse, Łukasz Kidziński.

**Writing – original draft:** Scott D. Uhlrich, Antoine Falisse.

**Writing – review & editing:** Scott D. Uhlrich, Antoine Falisse, Łukasz Kidziński, Julie Muccini, Michael Ko, Akshay S. Chaudhari, Jennifer L. Hicks, Scott L. Delp.

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
