## [Decision Letter · Decision Letter 0]

3 Jul 2023

Dear Dr. Uhlrich,

Thank you very much for submitting your manuscript "OpenCap: motor control and musculoskeletal forces from smartphone videos" for consideration at PLOS Computational Biology. As with all papers reviewed by the journal, your manuscript was reviewed by members of the editorial board and by several independent reviewers. The reviewers appreciated the attention to an important topic. Based on the reviews, we are likely to accept this manuscript for publication, providing that you modify the manuscript according to the review recommendations.

Sincerely,

Alison L. Marsden

Academic Editor

PLOS Computational Biology

Daniel Beard

Section Editor

PLOS Computational Biology

Reviewer's Responses to Questions

**Comments to the Authors:**

Reviewer #1: This is a well-written paper describing an innovative system with obvious applications in the clinic and in the "wild".

Please provide context for error values given in the validation section (lines 156-158, etc). For instance, do the MAE errors have the potential to affect clinical decision making?

Please discuss scatter in the predictive accuracy shown (e.g., Figures 3 and 4). Do errors in the accuracy of predicting peak MCF or knee extension moment have the potential to affect clinical decision making? Are there any indications as to the cause of larger errors in specific individuals / tests / motions?

Please discuss error relative to the capture volume. How does error relate to depth of field or position of the individual relative to the iPhone cameras or within the capture volume?

Please discuss repeatability of measurements.

Please briefly discuss practical guidance/requirements (e.g., clothing, lighting, other) for capturing iPhone video in the clinic or in the wild.

Reviewer #2: I would like to congratulate the authors of “OpenCap: motor control and musculoskeletal forces from smartphone videos” on an important and transformative contribution to the field. In this manuscript, the authors present, validate, and test OpenCap, an open-source platform for computing kinematics and kinetics of human movement using videos captured from smartphones. This is a transformative contribution to the field as it provides a tool for low-cost analysis of human movement, that can be easily administered in a clinical setting or practical environment, and can quickly output relevant metrics from a large cohort. The authors well validate the platform and test its utility for screening for disease risk, evaluating interventions, and informing rehabilitation. The manuscript is well written, all figures are of the highest quality, and all data is openly shared, with additional detailed descriptions in the supplemental material and elsewhere. I only have minor comments, which are provided below.

Minor comments:

Line 7: consider specifying “two or more smartphones” or “a minimum of two” instead of simply two

In lines 527-536 the authors describe how OpenCap can estimate dynamics using muscle driven tracking simulations of joint kinematics. However, it is unclear to me how the ground reaction forces are estimated and I would appreciate clarification on this point. I have referred to Appendix 1: Optimal control formulations to seek an answer. Potentially the GRFs were allowed to vary in the muscle driven tracking simulation and then computed based on the simulation that minimized the cost function specified in line 535? Given this statement (from OpenSim documentation) “Note that you must measure or model all external forces acting on a subject during the motion to calculate accurate muscle forces” it seems to me the GRFs must have been modeled somehow as they were not measured for the OpenCap condition.

Please comment on whether users can select a musculoskeletal model for use with OpenCap or whether OpenCap must use the model from Lai et al. with modified hip abductor muscle paths.

Lines 685-687: It is my understanding that the OpenCap muscle driven dynamic simulation is a forward dynamics approach, such that muscle activation dynamics are modeled. For the first application in assessing peak knee adduction moment and peak medial knee contact force during walking, OpenCap dynamic simulation is compared to MoCap static optimization (using OpenSim Static Optimization utility). Can the authors please comment on the choice of static optimization (which cannot account for muscle activation dynamics) here when a more similar comparison may have been to Mocap Computed Muscle Control driven forward simulation, which does account for muscle activation dynamics?

Lines 716-718 Same question as above regarding muscle driven tracking optimization versus static optimization but now in the context of squatting.

Line 381: Typo. Correct to read "inertial-measurement-unit-based" please.

**Have the authors made all data and (if applicable) computational code underlying the findings in their manuscript fully available?**

Reviewer #1: Yes

Reviewer #2: Yes

PLOS authors have the option to publish the peer review history of their article (what does this mean?). If published, this will include your full peer review and any attached files.

Reviewer #1: No

Reviewer #2: **Yes: **Carrie L Peterson

Figure Files:

Data Requirements:

Reproducibility:

References:

---

## [Decision Letter · Decision Letter 1]

24 Aug 2023

Dear Dr. Uhlrich,

We are pleased to inform you that your manuscript 'OpenCap: human movement dynamics from smartphone videos' has been provisionally accepted for publication in PLOS Computational Biology.

Best regards,

Alison L. Marsden

Academic Editor

PLOS Computational Biology

Daniel Beard

Section Editor

PLOS Computational Biology

Reviewer's Responses to Questions

**Comments to the Authors:**

Reviewer #1: Thank you for addressing my concerns and questions.

Reviewer #2: The authors have improved the manuscript to address all concerns/questions. I have no further concerns/questions.

**Have the authors made all data and (if applicable) computational code underlying the findings in their manuscript fully available?**

Reviewer #1: Yes

Reviewer #2: Yes

PLOS authors have the option to publish the peer review history of their article (what does this mean?). If published, this will include your full peer review and any attached files.

Reviewer #1: **Yes: **Todd L Bredbenner

Reviewer #2: **Yes: **Carrie L. Peterson

---

## [Editor Report · Acceptance letter]

12 Sep 2023

PCOMPBIOL-D-23-00154R1 

OpenCap: human movement dynamics from smartphone videos

Dear Dr Uhlrich,

I am pleased to inform you that your manuscript has been formally accepted for publication in PLOS Computational Biology. Your manuscript is now with our production department and you will be notified of the publication date in due course.

With kind regards,

Timea Kemeri-Szekernyes
